# DISSECTING AN ADVERSARIAL FRAMEWORK FOR INFORMATION RETRIEVAL

## ABSTRACT

Recent advances in Generative Adversarial Networks facilitated by improvements to the framework and successful application to various problems has resulted in extensions to multiple domains. IRGAN attempts to leverage the framework for Information-Retrieval (IR), a task that can be described as modeling the correct conditional probability distribution $p(d|q)$ over the documents ($d$), given the query ($q$). The work that proposes IRGAN claims that optimizing their minimax loss function will result in a generator which can learn the distribution, but their setup and baseline term steer the model away from an exact adversarial formulation, and this work attempts to point out certain inaccuracies in their formulation. Analyzing their loss curves gives insight into possible mistakes in the loss functions and better performance can be obtained by using the co-training like setup we propose, where two models are trained in a co-operative rather than an adversarial fashion.

## 1 INTRODUCTION

Information-Retrieval (IR) involves providing a list of ranked documents $\{d_1, d_2, \ldots, d_k\}$ in answer to a query $q$. This general formulation can be extended to various tasks like web-search, where the documents are web pages and information needs are queries, content-recommendation, where the documents are items/content to suggest and queries are users, and Question-Answering, where the documents are answers and queries are questions. The retrieved list can also be viewed as a probability distribution over candidates, one example being $Rank_q(d_i) \equiv p(d_i|q) \propto (\frac{1}{Rank_q(d_i)})^l$, where $l$ is a hyperparameter. Even if the probability distribution is not explicit, it is desirable to retrieve a higher ranked document more often than a lower ranked document.

GANs were proposed as alternatives to generative models and have been shown to be capable of modeling the true data well. High dimensional settings like images and word sequences have seen some success. Given that the generator in GANs tries to model the training data's distribution, adversarial setups seem like a natural fit for IR. The learned distribution can then be used to retrieve relevant documents for incoming queries. IRGAN is a framework proposed by Wang et al. (2017), with the hope of giving Information-Retrieval, access to the large literature of GANs.

IRGAN consists of a discriminator and a generator. Like in a typical setup, the discriminator learns to distinguish between documents produces by the real probability distribution or the real ranking and the generator's probability distribution. It increases the likelihood of the former and decreases it for the latter. The generator tries to bring its probability distribution closer to the real one so that it increases the likelihood of confusing the discriminator into believing that it is the true distribution. Ideally, equilibrium is achieved when the generator manages to rank the documents according to the true distribution.

However, the formulation and implementation of the loss function in the work seems to have a few issues. Specifically, the use of the baseline term recommended in the work results in pitting the loss functions of the discriminator and the generator directly against each other and this leads to issues that are conspicuous in the loss curves. The training starts off with a pre-trained discriminator and generator, and the performance of the generator decreases as the training proceeds, while you would actually expect the opposite. When pre-training is not used, the generator does not learn at all. This forces IRGAN to choose the generator or discriminator based on whichever has better performance, while it expected that the generator is chosen at equilibrium.

Given the traction this paper has received since its inception (53 citations as of $27^{th}$ September 2018), it is important to critically analyze the work and attribute the claimed performance improvements correctly. To this end, we propose two models which outperform IRGAN on two of the three tasks and give a comparable performance on the third. They also serve as an ablation study by experimentally showing that the generator might not be playing a vital role during train or test time.

The following contributions are made in this work

- We propose a model motivated by Co-training which outperforms IRGANs
- We point out inaccuracies in the minimax loss function used in IRGANs
- We substantiate the same by drawing conclusions from the loss curves

## 2    RELATED WORK

### 2.1    GENERATIVE ADVERSARIAL NETWORKS

Generative Adversarial Networks (GANs) (Goodfellow et al. (2014)) were proposed as an alternative to generative models (Salakhutdinov & Larochelle (2010)) which used Markov Chains or other approximations to compute intractable probability distributions. In essence, the *generator* tries to model the real data distribution and the *discriminator* learns to differentiate between real data points and generated data points. GANs are notoriously unstable to train and works like DCGANs (Radford et al. (2015)) and Wasserstein GAN (Arjovsky et al. (2017)) have successfully attempted to alleviate a few issues. Nonetheless, GANs have been widely applied to various problems like image generation, text generation, cross-modal retrieval and more niche ones like Interactive Image Generation (Zhu et al. (2016)), Text to Image (Zhang et al. (2017)), Image to Image style transfer (Isola et al. (2017)) and robotics (Bousmalis et al. (2017)).

While GANs allow generation based on a random variable $z$, Conditional GANs (Mirza & Osindero (2014)) partition the sample variable into two parts ($z$ and $y$). $y$ is used to denote which part of the probability distribution the generator has to generate from, and $z$ plays the same role played in Vanilla GANs (Goodfellow et al. (2014)). Conditional GANs dovetail with IR because $y$ can be used to represent the query or its embedding, and in theory, the model should be able to generate the required document.

$$y \sim query \qquad G(z|y) \sim p_\theta(d|z, q)$$

We feel that an eventual adversarial formulation for IR will be similar to this in flavor.

### 2.2    RETRIEVAL OF IMAGE RESPONSES

Creswell & Bharath (2016) employed Sketch-GANs for the interesting task of retrieving similar merchant seals (images) based on an input image. DCGANs (Radford et al. (2015)) are used to generate an image, and post training, the last layer of the discriminator is popped off and the rest of it is used as an encoder. This model, however, is specifically for retrieving image responses.

## 3    BACKGROUND

In the subsequent sections, $D$ denotes the discriminator, $G$ the generator, $p_{true}$ the real probability distribution over documents, $\phi$ the parameters of the discriminator, $\theta$ the parameters of the generator, $d$ the document, $q$ the query and $r$ the rank of a document with respect to a query.

The equations used to train the *discriminator* and *generator* in Goodfellow et al. (2014) are the following respectively.

$$\nabla_{\theta_d} \frac{1}{m} \sum_{i=1}^{m} \left[ \log D\left( \boldsymbol{x}^{(i)} \right) + \log \left( 1 - D\left( G\left( \boldsymbol{z}^{(i)} \right) \right) \right) \right]$$

$$\nabla_{\theta_g} \frac{1}{m} \sum_{i=1}^{m} \log \left( 1 - D\left( G\left( \boldsymbol{z}^{(i)} \right) \right) \right)$$

The discriminator minimizes the likelihood of a "generated" data point and maximizes it for a "real" data point, while the generator tries to generate data points which the discriminator thinks is "real". The two models are trained alternatively and the procedure culminates in a generator which is able to produce data which looks like the real data.

# 4 IRGAN FORMULATION

This section elucidates the IRGAN formulation (Wang et al. (2017)). Comments by the authors are in italics (in this section alone), while normal typeface is a paraphrased version of IRGAN. IRGAN is motivated by the combination of two schools of thoughts, the generative retrieval model and the discriminative retrieval model.

## 4.1 DISCRIMINATOR AND GENERATOR

The generative retrieval model $p_\theta(d|q, r)$ tries to sample relevant documents from a candidate pool with the aim of cloning the true probability distribution $p_{true}$. The discriminative retrieval model $f_\phi(q, d)$, which is a binary classifier, tries to discriminate between real and generated pairs $(q, d)$.

Two different loss functions IRGAN-Pointwise and IRGAN-Pairwise are proposed.

## 4.2 IRGAN-POINTWISE

This is called so because each data point is independently used to train, unlike in IRGAN-Pairwise where *pairs* of *points* are used. The dataset is expected to have some cue with respect to how often a document is correctly retrieved for a query, if at all.

$$J^{G^*, D^*} = \min_\theta \max_\phi \sum_{n=1}^{N} (E_{d \sim p_{true}(d|q_n, r)}[(\log D(d|q_n))] + E_{d \sim p_\theta(d|q_n, r)}[(\log 1 - D(d|q_n))])$$

Note that the generator $G$ can alternately be written as $p_\theta(d|q_n, r)$, which denotes the modeled probability distribution, and $D(d|q) = \sigma(f_\phi(d, q))$ represents the discriminator's score.

## 4.3 IRGAN-PAIRWISE

In some IR problems the training data may not be a set of relevant documents for each query, but rather a set of ordered document pairs $R_n = [< d_i, d_j > | d_i \succ d_j]$, where $d_i \succ d_j$ means that the first document is more relevant for query $q_n$ than the second document. $o$ represents a real pair $< d_u, d_v >$ and $o'$ represents a generated pair $< d'_u, d'_v >$. The discriminator's goal in this setting is to discriminate between $o$ and $o'$, with $D(o|q) = \sigma(f_\phi(d_u, q) - f_\phi(d_v, q))$

$$J^{G^*, D^*} = \min_\theta \max_\phi \sum_{n=1}^{N} (E_{o \sim p_{true}(o|q_n)}[(\log D(o|q_n))] + E_{o' \sim p_\theta(o'|q_n)}[(\log 1 - D(o'|q_n))])$$

Note the similarity between this and the previous formula. *The problem with this formula is that $D(o|q)$ is actually supposed to denote the probability that the pair $o$ is from the real data distribution and not the probability that the pair is correctly ranked, as mentioned in the paper.*

## 4.4 OPTIMIZING THE GENERATOR

The generator samples documents from the candidate pool based on its belief (relevance score). This sampling has the downside that the gradients cannot be backpropagated, and policy gradients (Sutton et al. (2000)) have to be used. *As an intuition, the documents can be considered as the arms of a contextual multi-arm bandit (Auer et al. (2002), Lu et al. (2010)), and picking an arm can be viewed as analogous to choosing the document as relevant. The policy discovered gives us the relevance of each document and $-\log(1 - D(d|q))$ is the reward for picking that action/document (d).* Let $J^G$ represent the objective function of the generator that it has to maximize. The policy gradient (REINFORCE) can be written as the following.

$$\nabla_\theta J^G(q_n) = \frac{1}{K} \sum_{k=1}^{K} \nabla_\theta \log p_\theta(d_k|q_n, r) \log(1 + exp(f_\phi(d_k, q_n)))$$

### 4.5 BASELINE TERM

To reduce the variance in REINFORCE, a standard trick is to use the advantage function instead of just the reward. This does not change the optimal parameters.

$$\log(1 + exp(f_\phi(d_k, q_n))) - \mathbb{E}_{p_\theta(d_k|q_n)}[\log(1 + exp(f_\phi(d_k, q_n)))]$$

Another baseline term that is suggested for each query is $f_\phi(d_+, q)$, where $d_+$ represents the positive document. This is legal because the term does not depend on the document (action). This is motivated by the belief of a larger generator score if $f_\phi(d_+, q)$ is large and lower if $f_\phi(d_+, q)$ is low. *This baseline term is used in two of their three tasks and causes the violation of adversarial formulation, as we show in the following section.*

## 5 INSIGHTS INTO IRGAN MINIMAX LOSS FUNCTION

Having shown that the generator can be optimized using REINFORCE, we focus on the loss function and show how the baseline term exacerbates training. We consider Stochastic Gradient Descent updates for ease of illustration. Consider a triple $(q, d_r, d_g)$, where $d_r$ denotes the correct document according to the true distribution and $d_g$ denotes the generated document. The discriminator's updates are in the direction of $\nabla J^D$, with the following definition.

$$J^D = \log D(d_r|q) + \log(1 - D(d_g|q))$$

With the baseline term included, the generator's updates are in the direction of $\nabla J^G$, with the following definition.

$$J^G = \log(1 - D(d_r|q)) + \log D(d_g|q)$$

Since maximizing $\log(1 - z)$ with respect to $z$ is the same as maximizing $-\log z$, we can write the following equivalent loss functions

$$J^D = \log D(d_r|q) - \log D(d_g|q) \qquad J^G = -\log D(d_r|q) + \log D(d_g|q)$$

Note that this substitution is similar in principle to the substitution in Goodfellow et al. (2014), where the motivation is to allow easier flow of gradients. It is apparent that the discriminator and the generator are optimizing directly opposite loss functions and this detrimental to the performance of the models. We provide experimental proof later that the performance improvements shown in IRGAN are mainly because of the discriminator maximizing the likelihood of the real data and not because of the generator.

## 6 PROPOSED MODELS

We propose two models to compare and critically analyze performance gains facilitated by IRGAN and illustrate them in Figure 1.

The first model increases the likelihood of the training data and decreases the likelihood of documents which are not relevant to the query but have a high score according to its *own* parameters. It maximizes the following, where the sampling for the second term is from a candidate pool with only negative answers (denoted by $p^-$). Not following this will lead to undesirable updates because sampling positive documents for the second term will result in decreasing the likelihood of real data. $\psi$ denotes the parameters of the only model.

$$J^{Model1} = \max_\psi \sum_{n=1}^{N} (E_{d \sim p_{true}(d|q_n, r)}[(\log D(d|q_n))] + E_{d \sim p_\psi^-(d|q_n, r)}[(\log 1 - D(d|q_n))])$$

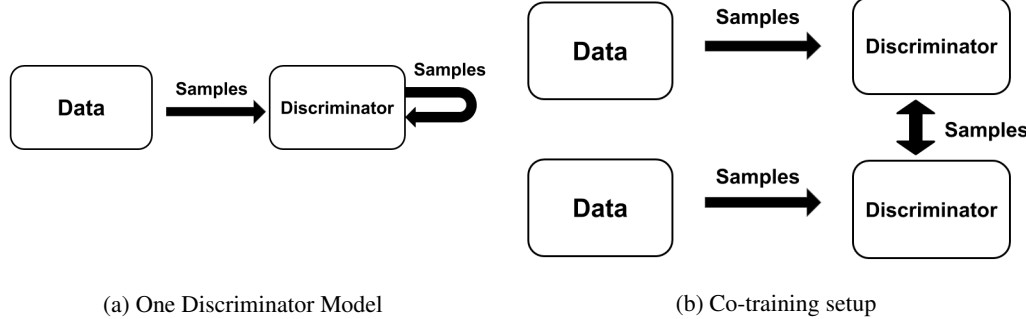

(a) One Discriminator Model  (b) Co-training setup

Figure 1: Models

To alleviate the pernicious loss function of IRGAN, we propose a model which uses two discriminators in a co-operative setup influenced by Co-training (Blum & Mitchell (1998)). Instead of using two different views $(x_1, x_2)$ as mentioned in the work, we use the same views for both the discriminators but let them influence each other in a feedback loop. Training is similar to Model 1 with the only difference being that each discriminator decreases the likelihood of documents relevant to the *other* discriminator rather than itself, as shown in the equation below. This model achieves better performance than IRGAN.

$$J^{Model1} = \max_{\psi_1} \sum_{n=1}^{N} (E_{d \sim p_{true}(d|q_n, r)}[(\log D(d|q_n))] + E_{d \sim p_{\psi_2}^-(d|q_n, r)}[(\log 1 - D(d|q_n))])$$

$$J^{Model2} = \max_{\psi_2} \sum_{n=1}^{N} (E_{d \sim p_{true}(d|q_n, r)}[(\log D(d|q_n))] + E_{d \sim p_{\psi_1}^-(d|q_n, r)}[(\log 1 - D(d|q_n))])$$

## 7 EXPERIMENTAL SETUP

This section describes the datasets, the task and hyperparameters.

### 7.1 DATASETS

We conduct experiments on three tasks, Web Search, Item Recommendation and Question Answering, using the same datasets mentioned in IRGAN.

Table 1: Datasets

| Task | Dataset |
|------|---------|
| Web Search | LETOR by Liu et al. (2007) |
| Item-Recommendation | Movielens |
| Question Answering | InsuranceQA by Feng et al. (2015) |

### 7.2 TASK

In Web Search, the task is to retrieve the document which is most relevant to the query. Each query on average has around 5 positive documents. In Content Recommendation, users give ratings for movies and given a user, the task is to retrieve a movie that they would probably rate high. In IRGAN, any movie retrieved for which the user rating is greater than or equal to 4 (out of a scale of 5) is considered correct. Based on the dataset statistics, around 55% of the user-movie ratings are $\geq 5$. This makes the problem easy to solve. In Question Answering, every query has just one relevant document in most cases. This is thus the hardest task.

### 7.3 HYPERPARAMETERS

The hyperparameters for the proposed model are the same except for absence of G_Epochs, for obvious reasons. Information about hyperparameter tuning is mentioned in the Appendix.

Table 2: Hyperparameters for IRGAN

| Hyperparameter | Description |
|---|---|
| Learning Rate | For both generator and discriminator |
| Batch Size | Batch size for training |
| Embed Dim | Embedding dimension of query or document |
| Epochs | Number of epochs of training |
| D_Epochs | Number of epochs the discriminator is trained per epoch |
| G_Epochs | Number of epochs the generator is trained per epoch |
| Temperature | Temperature parameter for softmax sampling of documents |

## 8 EXPERIMENTS AND DISCUSSION

We report only the P@5 and NDCG@5 values because all other metrics follow the same trend.

### 8.1 WEB SEARCH

Table 3 reports the performance of various models. As can be seen, both the Single Discriminator and the Co-training models outperform IRGAN models. The fact that each query is associated approximately with 5 positive documents provides evidence that the proposed models can perform well in sparse reward settings.

Table 3: Results on LETOR dataset used in IRGAN

| Model | P@5 | NDCG@5 |
|---|---|---|
| RankNet (Burges et al. (2005)) | 0.1219 | 0.1709 |
| LambdaRank (Burges et al. (2007)) | 0.1352 | 0.1920 |
| IRGAN-pointwise | 0.1657 | 0.2225 |
| IRGAN-pairwise | 0.1676 | 0.2154 |
| Single Discriminator | 0.1676 | 0.2190 |
| Co-training | **0.1733** | **0.2252** |

### 8.2 ITEM-RECOMMENDATION

This task, in contrast to the other two, has multiple relevant documents that can be retrieved for each query, making it slightly easier. Each user (query) rates a movie (document), and $55\%$ of the entries in the train set and $56\%$ in the test set are relevant pairs. It can be seen in Table 4 that the single discriminator model achieves only a slightly lower score, and given the small size of the dataset (943 users), it makes just 7 more mistakes when compared to IRGAN. This is not a statistically significant number, especially because the IRGAN generator is pre-initialized to a model which scores $0.34$ but our model learns from scratch.

### 8.3 QUESTION ANSWERING

After close correspondence with the authors of IRGAN, we obtained all the hyperparameters required for the models. Multiple random seeds were used in vain, the results in the paper for Question-Answering tasks could not be replicated. We instead mention the best results out of all random seeds. We believe that if there is some random seed which gives better performance for IR-GAN, it should do so for our model as well. The co-training model outperforms IRGAN-Pairwise.

Table 4: Results on Movielens Dataset

| Model | P@5 | NDCG@5 |
|---|---|---|
| BPR (Rendle et al. (2009)) | 0.3044 | 0.3245 |
| LambdaFM (Yuan et al. (2016)) | 0.3474 | 0.3749 |
| IRGAN-pointwise | **0.3750** | **0.4099** |
| Single Discriminator | 0.3675 | 0.3925 |
| Co-training | 0.345 | 0.373 |

Table 5: P@1 on InsuranceQA

| Model | P@1 |
|---|---|
| IRGAN-Pairwise | 0.616 |
| Single Discriminator | 0.614 |
| Co-training | **0.623** |

## 8.4 LOSS CURVES

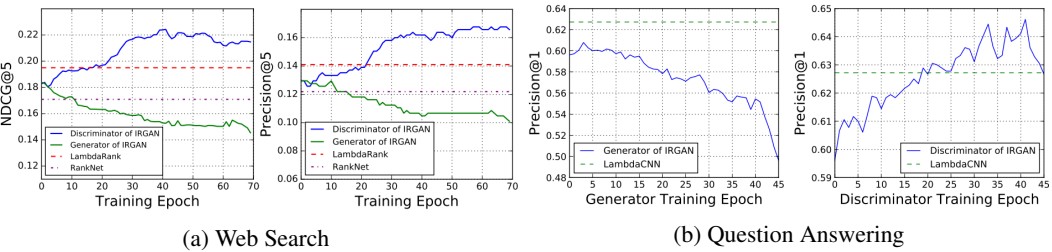

(a) Web Search

(b) Question Answering

Figure 2: Performance curves

The loss curves in Figure 2 picked from IRGAN's work show deteriorating performance of the generator, which is in contrast to what is observed in actual adversarial training. In the minimax setting, since the generator is expected to capture the real data distribution, its performance is supposed to improve and this can indirectly be seen in GANs and DCGANs where the samples generated look more and more like real-world data points. Further, a deteriorating generator implies that the discriminator's improvement in performance is only because of the first term of $J^D$, which hints that our proposed models might be able to do better than IRGAN. The reason offered in the paper is that "A worse generator could be the result of the sparsity of document distribution, i.e., each question usually has only one correct answer". But this reason does not seem plausible, given that DCGANs have been able to model very high dimensional data, where the probability distribution is only a tiny part of the real space.

Further, the increase in performance of the discriminator in all cases is coupled with a deteriorating generator. This substantiates our claim that the discriminator and the generator are optimizing directly opposite loss functions.

Item-recommendation task is a little different from the other two tasks at hand because of a large number of positive answers. When the loss curves are plotted, though the generator's performance improves, the discriminator's loss remains high and almost constant throughout the procedure, as shown in Figure 3. This is another indication that the performance of IRGAN is not actually because of the adversarial setup, but because of the maximization of the likelihood of the real data.

## 9 CONNECTIONS TO PREVIOUS WORK

We have already shown in Section 2 that Conditional GANs are connected directly to Information Retrieval. The problem can also be viewed as a contextual multi-armed bandit problem (Li et al.

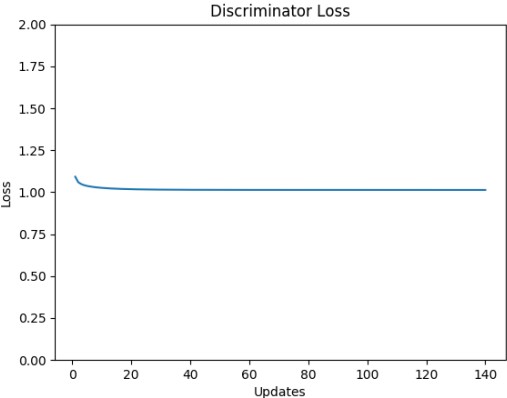

Figure 3: Discriminator Loss for Content-Recommendation

(2010)), where each documents is an arm and the context $x_{q,d}$ can be used to determine the action-value function $f_\theta(x_{q,d})$. In previous works (Li et al. (2010)) $f$ has been considered to be linear, but recent studies Collier & Llorens (2018) have modeled them as deep neural networks.

In Pfau & Vinyals (2016), a parallel is drawn between Actor-Critic algorithms (Konda & Tsitsiklis (2000)) and GANs. This is directly related to our work because REINFORCE (Sutton et al. (2000)) with a baseline can be connected to Actor-Critic algorithms when bootstrapping is used (Sutton & Barto (2018)). The work shows a restricted scenario which involves a stateless MDP, each action setting all the pixels of the image and cross-entropy loss instead of mean-squared Bellmann residual in which GANs are equivalent to Actor-Critic algorithms. But this equivalence holds only when the baseline term is not used so the formulation in IRGAN is not exactly equivalent to a GAN framework. Another study (Finn et al. (2016)) draws a parallel between Inverse Reinforcement Learning (Ng et al. (2000)) and GANs because both the methods try to "learn" the cost function to optimize for.

## 10    CONCLUSION AND FUTURE WORK

The experiments performed show that IRGAN is by no means state-of-the-art on those datasets. Further, the performance does not justify the large training time of $4$ hours per generator epoch and $1$ hour of discriminator epoch as opposed to $2$ hours per epoch of the co-training model (11 GB GPU and Question Answering task). The shaky mathematical formulation renders the generator useless after training, and any gains in performance can be attributed directly to the first term of $J^D$, where the likelihood of the real data is increased. We showed that the discriminator and generator are optimizing directly opposite loss functions and this is the cause of deleterious training.

The poor performance of IRGAN on Web-Search and Question Answering and only a satisfactory performance on Content-Recommendation (which has dense rewards) lead us to speculate that it does not work well in sparse reward scenarios. This is similar to a well-known problem called the Sparse Reward Reinforcement Learning. We think that a correct formulation along with established techniques from the former, like reward shaping (Ng et al. (1999)) may lead to better performance. Newer methods like Hindsight Experience Replay (Andrychowicz et al. (2017)) which allow models to learn both from mistakes and rewards may further ameliorate learning.

We would also like to explore in the direction of learning correct adversarial frameworks for more complex tasks like Image Retrieval and Question Answering which will involve learning end-to-end trainable models. With advances in modeling sequences, this could also involve generation of documents rather than sampling them.

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

## APPENDIX A    HYPERPARAMETERS

The hyperparameters are mentioned in tables 6, 7, 8, 9 and 10. The following were the ranges of hyperparameter tuning, along with the best value. Gradient Descent Optimizer was used so that the comparison with IRGAN is fair.

Table 6: Single Discriminator for Web-Search

| Hyperparameter/Seed | Range/List | Best |
|---|---|---|
| Learning Rate | 0.002-0.2 | 0.004 |
| Batch Size | [8,16,32] | 8 |
| Feature Size | [46, 92] | 46 |
| Random Seed | [20,40,60] | 40 |

For the co-training model, for every epoch, we optimize the two discriminators several times. We call these the outer and inner epochs in Table 7.

Table 7: Co-training for Web-Search

| Hyperparameter/Seed | Range/List | Best |
|---|---|---|
| Learning Rate | 0.002-0.2 | 0.006 |
| Outer Epochs | [30,50] | 50 |
| Inner Epochs | [30,50] | 30 |
| Batch Size | [8,16,32] | 8 |
| Feature Size | [46, 92] | 46 |
| Random Seed | [20,40,60] | 40 |

DNS_K in Table 8 represents the number of candidates that are chosen before performing the softmax. This is done to make the procedure computationally tractable. We use the value suggested in IRGAN.

Table 8: Single Discriminator for Content Recommendation

| Hyperparameter/Seed | Range/List | Best |
|---|---|---|
| Learning Rate | 0.01-0.05 | 0.02 |
| Batch Size | 10 | 10 |
| Embedding Dimension | [20, 40, 60] | 20 |
| Random Seed | 70 | 70 |
| DNS_K | 5 | 5 |

## APPENDIX B    DESCRIPTION OF MODELS

The discriminator and the generator have the same architecture in all the tasks. For the Web-retrieval task, the model has a single hidden layer $46$ units.

For the content-recommendation task, the model converts users and movies to a $5$ dimensional embedding. This can be though to be a single hidden layer which compresses a one-hot user embedding to a $5$ dimensional embedding.

For the Question-Answering task, each word is initialized to a $100$ dimensional random vector. A Convolutional Neural Network is then used and the window size of the convolutional kernel is (1,2,3,5). A max-pooling-over-time strategy is then used and the output is a $100$ dimensional vector because each feature map is pooled to a scalar. Note that this architecture is the same as the one used in the IRGAN paper. We refer the user to that for further description.

Table 9: Single Discriminator for Question Answering

| Hyperparameter | Best |
|---|---|
| Learning Rate | 0.05 |
| Epochs | 20 |
| Batch Size | 100 |
| Embedding Dimension | 100 |

Table 10: Co-training for Question Answering

| Hyperparameter | Best |
|---|---|
| Learning Rate | 0.05 |
| Outer Epochs | 20 |
| Inner Epochs | 1 |
| Batch Size | 100 |
| Embedding Dimension | 100 |

