# OpenReview forum: "Dissecting an Adversarial framework for Information Retrieval"
_ICLR.cc/2019/Conference_

### Official Review · AnonReviewer2 · 2018-10-31
**Good idea, but paper suffers on many important points**

**Rating:** 4
**Confidence:** 4

**Review:**

This paper trains an information retrieval (IR) model by contrasting the joint query-document distributions, p(q, d) with negative samples drawn from a resampling of the product of marginals, p(q) x p(d). They use a second discriminator to provide the re-weighting (I believe picking to top negative sample from the other model) and train this other model in a way that mirrors the first. They also attempt to point out some theoretical problems with a competing model, IRGAN, which uses a generator that is trying to model the joint.

While I like the proposal idea, I think the paper has too many problems to warrant publication. First, the story is very disappointing. The authors phrase most of the paper as a critique of IRGAN, but this critique falls short. Really this is more of a paper about where to get negative samples when training a model of the joint (or the log-ratio in this case). Using negative samples from real data with noise contrastive estimation [1] is found in numerous works in NLP [2][3], and has gained some recent attention in the context of representation learning [4][5]. The first algorithm proposed is essentially doing a sort of ranking loss on negative samples, which mirrors similar works [6]. In fact, the generator in IRGAN could be viewed as just a parametric / adaptive negative sampling distribution in the context of NCE for the ultimate purpose of learning an estimate of the log-ratio. The most interesting thing I think of this work here is the co-training, i.e., using another model to help re-sample, and I think this idea should be explored in more detail.

Second, the paper spends far too much time revisiting prior work than addressing their own model, doing more analysis, providing more insight.

Third, the paper is just poorly written. The notation is confusing, some of the equations are unclear (I have no idea how "r" is used in any of this), and the arguments of the baseline in IRGAN don't really doesn't make any sense.

Notes:
P1
I don't really follow why IRGAN is so central to this work. Good ideas aren't difficult to motivate, especially if empirically everything works out.
P2
I'm having trouble with claims, especially more recently, about GAN instability, particularly since numerous approaches [7][8] seem to have more or less solved the problem.

The use of "|" in G is awfully confusing.
P3
Almost 2 pages of unnecessary background

P4
Why are we using "|" in functions? What's wrong with ","?
theta = \theta
I don't understand the point of the quote (in italics).
What happened to "r" in all of this?
The last two equations and their relationship could be more clear.

You use italics, so is this supposed to be a quote? But then you have a section which attempts to show this.
P5
I have no idea what's supposed to be going on in 5). The samples from the real joint don't factor in the generator gradient, or at least it's absolutely not clear that this pops out of the baseline? Then you switch from log (1 - x) to - log x and there's some claim about this violating the adversarial objective?

It took me more than a few reads to figure out what the equation at the bottom of P5 is doing: is this resampling? It's fairly unclear.

[1] Gutmann, Michael U., and Aapo Hyvärinen. "Noise-contrastive estimation of unnormalized statistical models, with applications to natural image statistics."
[2] Mnih, Andriy, and Koray Kavukcuoglu. "Learning word embeddings efficiently with noise-contrastive estimation."
[3] Mikolov, Tomas, et al. "Distributed representations of words and phrases and their compositionality."
[4] Oord, Aaron van den, Yazhe Li, and Oriol Vinyals. "Representation learning with contrastive predictive coding."
[5] Hjelm, R. Devon, et al. "Learning deep representations by mutual information estimation and maximization."
[6] Faghri, Fartash, et al. "VSE++: Improving Visual-Semantic Embeddings with Hard Negatives."
[7] Miyato, Takeru, et al. "Spectral normalization for generative adversarial networks."
[8] Mescheder, Lars, Andreas Geiger, and Sebastian Nowozin. "Which Training Methods for GANs do actually Converge?."

---

> ### Author Response · Authors · 2018-11-27
> **Response**
>
> We thank the reviewer for going over the paper carefully and giving very useful feedback. We understand that there are issues the reviewer has with regards to the writing of the paper. We have clarified notation everywhere, reduced the unnecessary background material and added a section about what motivated us to choose the proposed model.
>
> The following summarizes the key idea of the paper, which the reviewer has related to noise constrastive estimation.
> We agree that it is important to generate the negative samples correctly. But unlike in [1], the generated negative samples depend on the query directly. In IRGAN, the generator’s score for each document paired with the given query is used Score(q,d). A document is sampled after normalizing this score and treating the normalized scores as a probability distribution. Therefore, P(d|q) \propto Score(q,d), where Score represents the generator’s score. In the co-training model that we proposed, instead of using the generator to sample negative data points, we use another discriminator. This makes the setup symmetric. However, this is not just negative sampling, but the two models run in a co-operative setup, with each model feeding in examples which are wrong to the other model. For an intuitive explanation, if Model 1 thinks with a high probability that Document 1 matches Query 1, which is wrong, then the pair (Document 1, Query 1) is fed as a negative sample to Model 2. In other words, query-document pairs which are matched wrongly by one model are fed to the other model. We have included better explanations in the paper as well.
>
> The following are paragraph-wise comments to explain the inaccuracies which we addressed and concerns that the reviewer has pointed out.
> 1)    IRGAN has gained a lot of traction recently. Though the formulation has some of the key ingredients of an adversarial framework for information retrieval, there are key issues we need to answer before we can use it. This paper acknowledges the good parts of IRGAN while pointing out potential loopholes and improvements to the same.
> 2)    We understand that the comment about the instability of GANs was a little unnecessary, and we have removed it. The use of “|” in G is borrowed from [2] which is the Conditional GANs paper, we apologize for the confusion.
> 3)    We agree that there was some unnecessary related work and background work. We feel that IRGAN’s framework is important to understand and have retained that, but we have trimmed down the sections so it is easier to follow now.
> 4)    We are sorry that the use of the pipe has led to some confusion and added a comment about it. We wanted to stick to the exact equations used in IRGAN. D(q|d), as you pointed out, is notationally similar to D(q,d). The italics are mainly being used to comment on the IRGAN’s framework. We wanted to make a distinction between IRGAN paper’s comments and our comments. All the text without the italics is a paraphrased version of IRGAN. “r” is used to denote the rank of the document with respect to that query and is borrowed directly from the IRGAN paper. It can be ignored and the details of the formula don’t change. But we wanted to use the exact same equations. We have modified them for easier understanding though.
> 5)    The crux of the section the reviewer is talking about is that the loss functions that the discriminator and generator are optimizing are exactly opposite. Even though the real joint does not factor in the generator’s gradient, it is important to see that the sign of the gradient is determined by it, which is in turn determined by the discriminator. REINFORCE algorithms are very sensitive to the sign of the rewards, and this can be seen on Page 15 in [3].
>
> Apart from this, we have made efforts to make the paper more readable. We have also added a section which connects the framework to Conditional GANs, Contextual Multi-Armed Bandits and Actor-Critic Algorithm.
>
> We thank the reviewer for a careful perusal of the paper and value the feedback given. We sincerely hope that the reviewer modifies the score based on our revised submission. We have also added a section which connects the framework to Conditional GANs, Contextual Multi-Armed Bandits and Actor-Critic Algorithm.
>
> We thank the reviewer for a careful perusal of the paper and value the feedback given. We sincerely hope that the paper is easier to understand now.
>
> [1] - Mikolov, Tomas, et al. "Distributed representations of words and phrases and their compositionality." Advances in neural information processing systems. 2013.
> [2] - Mirza, Mehdi, and Simon Osindero. "Conditional generative adversarial nets." arXiv preprint arXiv:1411.1784 (2014).
> [3] - http://rail.eecs.berkeley.edu/deeprlcourse/static/slides/lec-5.pdf

---

### Official Review · AnonReviewer3 · 2018-11-01
**This paper points out the limitation of IRGAN but it  is better to submit to SIGIR.**

**Rating:** 5
**Confidence:** 3

**Review:**

This paper is closely related to the SIGIR 2017 paper “IRGAN: A minimax game for unifying generative and discriminative information retrieval models”. The SIGIR paper proposed a Generative Adversarial Nets (GAN) model for Information Retrieval (IRGAN). And in this paper, the authors dissect the IRGAN model and figure out that the discriminator and generator of IRGAN are optimizing directly opposite loss functions. They also provide experimental studies and show that the superiority of IRGAN in the experiments are mainly because of the discriminator maximizing the likelihood of the real data and not because of the generator.
Strong Points:
Considering that the IRGAN becomes popular after it being published, I should say the analyzing of this paper is important for researchers in the IR domain.

Concerns or Suggestions:
1.	Except the analyzing on IRGAN, the contribution of this paper is limited. Most of the parts of this paper introduce GAN and IRGAN. Only Section 5 focuses on the analyzing. The methods claimed new proposed, Single Discriminator and Co-training, are good for supporting the analyzing but they are not quite novel.
2.	It is strange to introduce the two models, Single Discriminator and Co-training in the experimental setting section. I would suggest to separate them out and introduce them earlier.
3.	The topic of this paper is more related to the IR domain. It will be better to publish it in SIGIR, together with the IRGAN paper.
4.	Besides, if it is possible, I would suggest researchers who have direct experiences on the implement and study on IRGAN give more comments on this paper.

---

> ### Author Response · Authors · 2018-11-27
> **Response**
>
> We thank you for carefully reading and understanding the crux of the paper. Indeed, we feel that IRGAN needs to be further studied before it is used in practice, and the main aim of the paper has been to point out loopholes in the formulation and motivate the proposed co-training model. We have made a few revisions to the paper after taking your suggestions into consideration and we hope they sufficiently address your concerns.
>
> 1) We have made the notation clearer in the equations and hope that they can be easily understood now.
> 2) We have added a separate section with the proposed models as suggested. We have also added what motivated us to choose those models.
> 3) Though we agree that IRGAN focusses on Information-Retrieval, we feel that the framework itself is more general and can be applied to tasks like content recommendation and Question Answering. In the future, the same framework could also be used for natural language generation (of documents) or image generation (to address the query needs). For us, the application in dialogue generation is the most interesting and we see great potential in this method.
> 4) We have added a section linking IRGAN to other works like Conditional GANs, Contextual Multi-armed bandits and actor-critic algorithms. We feel that these, especially the last one, will help make intuitive connections to the problem at hand.
> We hope the paper is easier to follow now. We thank you for your valuable feedback and time.

---

### Official Review · AnonReviewer1 · 2018-11-06
**Interesting findings though depth and rigor could have been better**

**Rating:** 6
**Confidence:** 4

**Review:**

This paper tries to argue that the formulation of IRGAN (a method from 2017 that aimed to use GANs for the standard IR task of estimation query-document relevance) is now well-founded and has inherent weaknesses. Specifically the paper claims that (unlike regular GANs and what was likely intended by the authors or IRGAN) the discriminator and generator are working against each other. The paper then aims to show a couple of more well-founded different (generator-free) setups that perform about as well (if not better) as the original IRGAN work.

Overall I found the work to be quite interesting and the findings to be illuminating. That said I think the paper notably lacked rigor and depth which definitely hurt the quality of the paper.

Below are my thoughts on the different facets as well as more detailed strengths / weaknesses breakdown:

Quality: Above average
As mentioned I think some of the findings are illuminating and thus overage the paper scores well on this aspect.

Clarity: Slightly above average
While the paper is largely easy to follow, there are certain key sections that are not well explained / have fundamental errors.

Originality: Strong

Significance: Little below average
My (main) concern here with this work is that the missing rigor and depth of the work is what is needed for readers to have a deeper understanding of the fundamental issue so as to be able to rectify it in future works.

---

Strengths / Things I liked about the work:

+ The topic / theme of the work: I believe as a community we should encourage more such works that take a critical deep dive into recently proposed methods that may have some inherent weaknesses. As the authors noted the IRGAN work has become quite popular despite some of these (previously unknown) issues.

+ The experimental results in general do a fair job illustrating the likely issue (though I would have liked to see more rigor and depth here as well as detailed below)

Weaknesses / Things that concerned me:

- (W1) Lacking rigor / depth: One of my big concerns with this work is that the analysis to demonstrate the inherent flaws of IRGAN if fairly shallow and not detailed enough. For example, Section 5 (which should have been the key section of the work) is quite poorly written and not rigorous enough. Claiming that log(1-z) can be replaced with - log(z) is incorrect -- how can this substitution be made as is?

Overall my sense after reading the work is that I understand that the IRGAN formulation is not completely well-formed in terms of discriminator/generator synergy (the pairwise formulation has the additional issue of separating real pairs rather than higher rank-lower rank pairs). However I do not buy that the generator and discriminator directly oppose each other as is claimed in the work (I believe this arises only due to the incorrect claim that log(1-z) can be replaced with -log(z)).

Thus at the end of the day I feel the reader is willing to buy there is a issue with the formulation, but they do not fully understand it not do they understand enough to understand how to rectify the underlying issues. To me that was unfortunate as the paper would have been an excellent work if it had done so.

- (W2) Missing some experimental results / deeper insights : There were some notable empirical results that were missing or not provided, that raised some concerns in my mind. For instance I don't see the co-training approach listed for the MovieLens dataset .. Why so? The authors make a secondary claim that they are able to improve upon IRGANs via their proposed approach but then they do not substantiate these on all the datasets which seems like a notable oversight.

 - (W3) Missing details: To add to the above I think the authors can clearly be more detailed in describing for instance the models for D, G, p_\psi etc .. Right now I can speculate what they are but I don't think a reader should be expected to speculate in such cases. Likewise empirical details about the datasets and their sizes could easily have been added.

Also the paper presents the IRGAN pairwise approach and mentions pairs in a couple of places but I don't see an approach that can learn from pairs among the ones proposed.

Another example is the two proposed models Fig 2a (Only discriminator) vs Fig 2b (cotraining). I don't see an explanation or intuition for why 2b is expected to be better than 2a. Given the claims of the work I would have wanted to understand this better.

- (W4) Significance testing: This is an important experimental process to understand the validity of some of the claims. While I understand it is not the main claim of the paper, understand the significance of these differences helps put things in perspective. I would strongly urge the authors to add this for all of their experimental results not just the ones the proposed models are outperformed by IRGANs.

- Lastly I would urge the authors to be rigid and clear in their notations. For example in the equation in section 4.5, "o'" occurs out of nowhere.

---

> ### Author Response · Authors · 2018-11-27
> **Response**
>
> We thank the reviewer for carefully reading the paper and understanding the crux even though our writing was not very clear in a few sections. As pointed out by the reviewer, the main motivation of the paper was to point out a few loopholes in IRGAN, which led to the proposal of a co-training based setup. It further shows that co-operative setups might perform as well, if not better, than adversarial setups. The following are a few comments which we hope to address the reviewers concerns.
>
> 1)    W1 – We understand why the reviewer feels there is a lack of rigor. But the substitution in point is very similar to the substitution made in the original GANs paper to allow easier flow of gradient. The substitution results in equivalent loss functions because the optimal value does not change, though the convergence speed etc might change. But we are theoretically motivating the substitution. We agree we should have been clearer about it and we have added a comment about it in the section. Log(1-x) is maximized when x=0, and -log(x) is maximized when x=0. This can be seen graphically at this link [https://www.wolframalpha.com/input/?i=plot+log(1-x)+and+-log(x) ].
> 2)    W2 – We are sorry that the reviewer felt there were missing details in the paper. We did not have results for the co-training model at that time, and we have run the experiments now and have added the results.
> 3)    We agree that the details about the model and parameters are not very clear. We have added an Appendix to help clear up a few doubts about the same.
>
> An intuitive explanation for why the co-training model does better than single discriminator is that it decorrelates the mistakes that the models are making. The mistakes made by one model are fed into another model, instead of self-feedback.
>
> Apart from this, we have made efforts to make the paper more readable. We have also added a section which connects the framework to Conditional GANs, Contextual Multi-Armed Bandits and Actor-Critic Algorithm.
>
> We thank the reviewer for such informative comments and feedback and hope that our revised version addresses some concerns and is more readable.

---

### Meta-Review · Area_Chair1 · 2018-12-15
**An interesting contribution, but lacking in depth.**

**Confidence:** 4
**Recommendation:** Reject

**Metareview:**

The manuscript centers on a critique of IRGAN, a recently proposed extension of GANs to the information retrieval setting, and introduces a competing procedure.

Reviewers found the findings and the proposed alternative to be interesting and in one case described the findings as "illuminating", but were overall unsatisfied with the depth of the analysis, and in more than one case complained that too much of the manuscript is spent reviewing IRGAN, with not enough emphasis and detailed investigation of the paper's own contribution. Notational issues, certain gaps in the related work and experiments were addressed in a revision but the paper still reads as spending a bit too much time on background relative to the contributions. Two reviewers seemed to agree that IRGAN's significance made at least some of the focus on it justifiable, but one remarked that SIGIR may be a better venue for this line of work (the AC doesn't necessarily agree).

Given the nature of the changes and the status of the manuscript following revision, it does seem like a more comprehensive rewrite and reframing would be necessary to truly satisfy all reviewer concerns. I therefore recommend against acceptance at this point in time.